# Beta-Lactam Antibiotic Resistance Genes in the Microbiome of the Public Transport System of Quito, Ecuador

**DOI:** 10.3390/ijerph20031900

**Published:** 2023-01-20

**Authors:** Fernanda Hernández-Alomía, Carlos Bastidas-Caldes, Isabel Ballesteros, Gabriela N. Tenea, Pablo Jarrín-V., C. Alfonso Molina, Pablo Castillejo

**Affiliations:** 1Grupo de Investigación en Biodiversidad, Medio Ambiente y Salud (BIOMAS), Universidad de Las Américas, Quito 170125, Ecuador; 2One Health Research Group, Universidad de las Américas, Quito 170125, Ecuador; 3Programa de Doctorado en Salud Pública y Animal, Universidad de Extremadura, 10003 Cáceres, Spain; 4Departamento de Genética, Fisiología y Microbiología, Facultad de Biología, Universidad Complutense de Madrid, 28040 Madrid, Spain; 5Biofood and Nutraceutics Research and Development Group, Faculty of Engineering in Agricultural and Environmental Sciences, Universidad Técnica del Norte, Ibarra 100150, Ecuador; 6Dirección de Innovación, Instituto Nacional de Biodiversidad, Quito 170525, Ecuador; 7Instituto de Investigación en Zoonosis (CIZ), Universidad Central del Ecuador, Quito 170521, Ecuador; 8Facultad de Medicina Veterinaria y Zootecnia, Universidad Central del Ecuador, Quito 170521, Ecuador

**Keywords:** m*ecA*, *bla*
_TEM-1_, *bla*
_OXA-181_, *bla*
_CTX-M-1_, environmental-DNA, antibiotic-resistance

## Abstract

Multidrug-resistant bacteria present resistance mechanisms against β-lactam antibiotics, such as Extended-Spectrum Beta-lactamases (ESBL) and Metallo-β-lactamases enzymes (MBLs) which are operon encoded in Gram-negative species. Likewise, Gram-positive bacteria have evolved other mechanisms through *mec* genes, which encode modified penicillin-binding proteins (PBP2). This study aimed to determine the presence and spread of β-lactam antibiotic resistance genes and the microbiome circulating in Quito’s Public Transport (QTP). A total of 29 station turnstiles were swabbed to extract the surface environmental DNA. PCRs were performed to detect the presence of 13 antibiotic resistance genes and to identify and to amplify 16S rDNA for barcoding, followed by clone analysis, Sanger sequencing, and BLAST search. ESBL genes *bla*_TEM-1_ and *bla*_CTX-M-1_ and MBL genes *bla*_OXA-181_ and *mecA* were detected along QPT stations, blaTEM being the most widely spread. Two subvariants were found for *bla*_TEM-1_, *bla*_CTX-M-1_, and *bla*_OXA-181_. Almost half of the circulating bacteria found at QPT stations were common human microbiota species, including those classified by the WHO as pathogens of critical and high-priority surveillance. β-lactam antibiotic resistance genes are prevalent throughout QPT. This is the first report of *bla*_OXA-181_ in environmental samples in Ecuador. Moreover, we detected a new putative variant of this gene. Some commensal coagulase-negative bacteria may have a role as *mecA* resistance reservoirs.

## 1. Introduction 

Bacterial diseases have an enormous impact on human health and remain a major focus in modern medicine. However, there is ample evidence that multidrug resistance (MDR) in pathogenic microorganisms, defined as showing resistance to three or more classes of antibiotics, has become a serious problem in the mismanagement of infectious diseases [1]. Given that infections are easily transmitted from person to person, microbes in public places, such as transport systems, can be a serious health problem [2].

The most widely used class of human antibacterial agents are β-lactams, which target the transpeptidases responsible for cross-linking peptidoglycan in cell walls, thereby inhibiting cell wall biosynthesis. There are more than 34 β-lactam antibiotics approved by the US Food and Drug Administration, which together account for approximately 50% of all antibiotic prescriptions worldwide and up to 60% in Ecuador [2,3,4]. However, to date, bacteria have acquired resistance mechanisms to overcome all major classes of β-lactam antibiotics. 

In Gram-negative pathogens, the main clinically determining mechanism of β-lactam resistance is the enzymatic inactivation of the antibiotics by β-lactamases [4]. Extended-spectrum β-lactamase (ESBL) is the group of β-lactamase enzymes that provides widespread resistance to β-lactam antibiotics, including penicillin, cephalosporins, and monobactam, thus, affecting the effectiveness of all beta-lactams but not carbapenems. These enzymes stimulate hydrolysis of the β-lactam ring and thereby inhibit these antibiotics [5]. Among ESBLs, the most widespread and clinically relevant are the class Temoneira (TEM), Sulfhydril Variable (SHV), and Cephotaxime Munich (CTX-M) types. ESBLs are frequently plasmid-encoded. Plasmids responsible for ESBL production frequently carry operon genes encoding resistance to other drug classes (for example, aminoglycosides or even polymixins) [6]. Therefore, antibiotic options in the treatment of ESBL-producing organisms are extremely limited [7]. Furthermore, carbapenems evade most β-lactamases but are hydrolyzed by metallo β-lactamases (MBLs) such as New Delhi metallo β-lactamase (NDM), *Klebsiella pneumoniae* carbapenemase (KPC), and OXA [8]. Thus, these enzymes are also named carbapenemases. 

Nevertheless, Gram-positive bacteria have evolved other mechanisms to avoid β-lactam inhibition of cell wall biosynthesis. Instead of cleaving the β-lactam ring, these bacteria alter penicillin-binding proteins (PBPs) through successive mutations. As a result, the β-lactams are less effective at disrupting cell wall synthesis. Methicillin-Resistant *Staphylococcus aureus* (MRSA) is one of the most widespread types of antibiotic resistance worldwide in Gram-positive pathogens [9]. This notorious spread is possible due to the presence of the staphylococcal chromosomal cassette *mec* (SCCmec) and plasmid-mediated genes, such as *blaZ*. These genetic elements work as the main drivers of β-lactam resistance in MRSA. Sharing homologous architecture with the *bla* operon, SCCmec contains the *mecA* gene, which encodes a modified penicillin-binding protein, PBP2a, with reduced affinity for β-lactam antibiotics [10]. Although *mecA* is the predominant variant, a divergent *mecA* homolog, *mecC*, was identified in MRSA strains from human samples in Ireland in 2011 [11]. Later, two new *mec* genes were reported, *mecB* and *mecD*, which are much less frequent and were both detected in *Macrococcus caseolyticus* [12,13].

All these resistance mechanisms have been typically detected from nosocomial environments. However, there is more evidence that the main selection mechanisms of bacterial resistance are increasing in non-hospital settings, having a direct influence as reservoirs of resistant pathogens and their spread capacity [14]. In 2015, the World Health Organization (WHO) implemented “the Global Action Plan on Antimicrobial Resistance” to encourage the monitoring and research on bacterial antibiotic resistance by governments and city councils [15]. Thus, the universal collection of data on microbial communities and horizontal-transfer systems contributed to the development of novel health and ecological surveillance programs in cities [16,17]. 

One of the strategies for MDR surveillance is the detection of antibiotic resistance genes in pathogenic and non-pathogenic microorganisms by amplifying and sequencing specific genes directly from environmental DNA (eDNA). Among the multitude of urban environments, transportation systems represent a uniquely centralized place. Therefore, a global collection of information on microbial communities of transportation systems has been gathered to contribute to this goal [16,17]. Specifically, Quito’s Public Transport (QPT) generates 2.9 million daily trips and it is where urban and rural dwellers meet and circulate daily. However, records of β-lactam resistance genes circulating in the QTP have been absent before the present publication. 

Given the widespread concern and lack of information on the risks for bacterial infections in public transportation, the present study aimed at surveying the main β-lactam antimicrobial resistance genes associated with QPT. Thus, we screened the presence of *mecA*, ESBL, and MBL resistance genes and the circulating bacteria by 16S rDNA genetic metabarcodes from the environmental DNA (eDNA) isolated from QPT surfaces. 

## 2. Materials and Methods 

### 2.1. Study Area and Sample Collection 

This was a cross-sectional study using primary data conducted in QPT. Quito is the capital of Ecuador and the second most populous city after Guayaquil, with 3.5 million inhabitants [18]. QTP comprises three main routes that cross the city from north to south. Bacteria were collected as described by Rawlinson et al. [19] by rubbering three sterilized cotton swabs (previously soaked in PBS 1X, pH 7.4) on the entire facing surface (ca. 20 cm^2^) of entry and exit turnstiles from 29 stations The three swabs from the same station were merged in 1.5 mL microtube obtaining 29 samples for further DNA extraction. 

### 2.2. eDNA Extraction and Gene Amplification

eDNA was directly extracted from the cotton swabs using the PureLink^TM^ Microbiome DNA Extraction Kit (#A29790 Fisher Scientific, Waltham, MA, USA) and concentrated with GeneVac^TM^ miVac Centrifugal Concentrator (Fisher Scientific, Waltham, MA, USA). eDNA concentration and purity were determined in NanoDrop™ at 230, 260, and 280 nm (Fisher Scientific, Waltham, MA, USA). The amplification of resistance genes was performed with specific primers for *bla*_TEM_, *bla*_SHV_, *bla*_CTXM-1_, *bla*_CTXM-2_, *bla*_CTXM-9_, *bla*_CTXM-8/25_, *bla*_NDM_, *bla*_KPC_, *bla*_VIM_, *bla*_OXA-48/181_, and *mecA.* The 16S rDNA V1-V3 region was amplified for bacterial identification. Amplification was performed in reactions of 15 µL containing 2X GoTaq^®^ Green Master Mix (Promega), 0.3 µM of each primer, and 0.1 to 0.5 ng/µL of extracted DNA. PCR was performed following the conditions described in Table 1. The positive control for ESBLs and carbapenemases was kindly donated by the Osaka Institute of Public Health, Osaka, Japan [20]; The positive control for *mecA* was obtained from a previous study on medical students-MRSA carriers in Ecuador [21].

### 2.3. 16S rDNA Clone Library 

16S rDNA V1-V3 amplicons were inserted onto a pCRTM4-TOPO^®^ TA vector, following TOPO™ TA Cloning™ Kit for Sequencing protocol (#K4575J10 Fisher Scientific, Waltham, MA, USA). Chemically competent *E. coli* DH5a cell preparation and transformation protocol were performed, as stated by Hanahan in 1983 [25]. X-gal and IPTG were used for the selection of positive colonies. Plasmids were extracted by alkaline lysis [26].

### 2.4. Sequence Analysis

PCR products for β-lactam resistance and 16S rDNA clones were sequenced by the Sanger technique in an ABI 3500xL Genetic Analyzer (Applied Biosystems, Foster City, CA, USA) by BigDye 3.1^®^ capillary electrophoresis matrix. Sequences were edited with MEGA X software [27] and compared against the GenBank database at the National Center for Biotechnology Information (NCBI) using the Basic Local Alignment Search Tool (BLAST). Sequences were submitted to NCBI for obtaining the corresponding accession numbers (see Section “Data Availability Statement”).

Chimeric sequences were removed with UCHIME2 Algorithm [28]. Kraken2 was used for the taxonomic assignment of the obtained sequences. Kraken2 was used for the taxonomic assignment of the obtained sequences. A Krona Pie Chart was used for the visualization of the taxonomic profile [29,30,31]. 

## 3. Results

### 3.1. QTP β-Lactam Resistance Genes Screening 

A total of 29 stations were sampled from the three main lines of QPT, distributed as shown in Figure 1. Four out of eleven genes for β-lactam antibiotic resistance were detected: *bla*_TEM_, *mecA*, *bla*_CTX-M-1_, and *bla*_OXA181_. The most widely spread resistance gene was *bla*_TEM_, detected in 26 stations, followed by *mecA* in 16 stations, *bla*_CTX-M-1_ in three, and *bla*_OXA181_ in two. Meanwhile, *bla*_SHV_, *bla*_CTX-M2_, *bla*_CTX-M8/25_, *bla*_CTX-M9_, *bla*_KPC_, *bla*_VIM_, and *bla*_NDM_, did not amplify in any sample (Figure 1; Appendix A). 

The sequencing of the PCR products for *bla*_TEM_ showed two subvariants named TV1 and TV2 with one nucleotide difference that leads to a single amino acid change from Ala116Val. These subvariants were assigned the accession numbers OP846058 and OP846059, respectively. Both variants matched the *bla*_TEM-1_ variant in the GenBank database. The variant TV1 showed 100% identity with *bla*_TEM-1_ for synthetic plasmids and species from *Staphylococcus*, *Streptococcus*, and *Pseudomonas* genera. TV2 was 100% identical to *bla*_TEM-1_ from *Klebsiella pneumoniae*, *E. coli*, and *Enterobacter cloacae* (Figure 1; Appendix A). 

Likewise, two *bla*_CTX-M-1_ subvariants were sequenced and named CV1 (accession number OP846056) and CV2 (accession number OP846057), which differentiate by a single amino acid change from Ala56Val. Both variants were found with 100% identity in *Escherichia coli*, *Klebsiella pneumoniae*, *Shigella*, and *Enterobacter*. Additionally, CV2 was also found in *Salmonella enterica* and *Vibrio* (Figure 1; Appendix A). 

The two subvariants identified for *bla*_OXA-181_ were called OV1 (accession number OP846061) and OV2 (accession number OP846062), which differ by a single amino acid change from His174Arg. The OV2 variant had 100% identity with *E. coli*, *K. pneumoniae*, *Enterobacter*, *Pseudomonas*, and *Shewanella*. Just one 100% match was found for OV1, which belonged to a sample of *E. coli* that was isolated from urine samples [32]. The other matches were the same as OV2, with 99% similarity.

### 3.2. QTP Microbiome

We examined QPT microbiome through a total of 293 clones of 16S rDNA partial gene amplification from eDNA. After data curation, around 275 sequences were obtained, and 256 were filtered by Kraken2 for taxonomic analysis (Figure 2). All the detected sequences belonged to microorganisms in the Bacteria domain, except for three belonging to the chloroplasts in *Vicia faba* (broad bean), *Colquhounia coccinea* (mint), and *Klebsormidium flaccidum* (algae) (Appendix A). The results detailed in Figure 2 showed that species-level identification was achieved for 74% of the sequences, which had over 97% similarity with sequences at the GenBank after the BLAST search (Appendix A). 

Almost half of the recorded 16S amplicons (49%) matched species in the Gram-negative phylum Proteobacteria. Species in Proteobacteria were distributed in the Alpha and Gammaproteobacteria classes (20% each), followed by the Betaproteobacteria class (8%) and Delta/epsilon subdivision (0.8%). The rest of the Gram-negative species belonged to the Bacteroidetes (4%) and Fusobacterium (1%) phylla. The most abundant phyla for gram-positive species were Firmicutes and Actinobacteria, with 21% and 20% of the abundance, respectively. Within Firmicutes, the Bacilli class was dominant, while within Actinobacteria, the dominant class was Actinobacteria. The Clostridiales order showed a relative abundance of only 3%. 

In the Gammaproteobacteria class, 15% of recorded sequences corresponded to the Enterobacteriaceae family, with enteric microbiota species such as *E. coli*, *Salmonella enterica*, *Klebsiella* sp., *Erwinia* sp., and common plant microbiota such as *Pantoea calida*, *P. aglomerans*, and *P. vagans*. Environmental species in the Gammaproteobacteria class were represented by the Pseudomonadales order (4%), including *Pseudomonas psychrotolerans*, *P. putida*, *P. stutzeri*, *P. brassicacearum*, *P. mendocina*, *Acinetobacter johnsonii*, and *A. radioresistens* in order of abundance (Appendix A). 

The Bacilli class included the *Staphylococcus* genus as the most abundant, including 9% of the 16S sequences with ten identified species: *S. saprophyticus*, *S. epidermidis*, *S. simulans*, *S. aureus*, *S. lugdunensis*, *S. capitis*, *S. pettenkoferi*, *S. simiae*, *S. xylosus*, and *S. haemolyticus* in order of abundance. The *Streptococcus* genus also showed a wide diversity, with important species such as *S. pneumoniae*, *S. mitis*, *S. dentisani*, *S. thermophilus*, and *S. cristatus* (Appendix A). 

In the Alphaproteobacteria class, the most abundant genus was *Mesorhizobium*, with 15% of the sequences (Appendix A). Finally, the Actinobacteria class was represented by common skin microbiota such as *Cutibacterium acnes* with 7% of the sequences. Other typically saprophytic and mammal-skin species, such as *Micrococcus luteus* and three species from the *Corynebacterium* genus, were also detected (Appendix A). 

In brief, 41.2% of the recorded 16S sequences were putative human microbiota: 29.7% from the skin, 18.7% from the respiratory tract, 9% from the gastrointestinal tract, 14.8% from the oral cavity, and 6.5% from other surfaces. The remaining species belonged to characteristically environmental bacteria, including one cyanobacterium and several chloroplasts from broad beans, flowers, and green algae (Appendix A). 52% of the analyzed sequences were putative pathogens. 

## 4. Discussions

To our knowledge, this is the first study of β-lactam resistance genes and the microbiome in a public transport system in Ecuador. Our results showed the prevalence of ESBL, MBLs, and *mecA* resistance genes along QPT. The presence of these genes agrees with previous environmental studies in Ecuador, where ESBL pathogens [33] and methicillin resistance in community-acquired infections [21] were reported. Two subvariants were found for *bla*_TEM-1_, *bla*_CTX-M-1_, and *bla*_OXA-181_, respectively. The *bla*_TEM-1__TV2 subvariant has been widely reported in three leading pathogens for deaths attributed to multidrug-resistant (MDR) bacteria, which are *Acinetobacter baumanii* [34,35], *E. coli* [36], and *K. pneumoniae* [37,38]. Nevertheless, most of the *bla*_TEM-1__TV1 subvariant matches at the GenBank were sequences related to synthetic plasmids. One of these was used for experiments on the evolution of β-lactamase, where the antibiotic resistance protein TEM-1 evolved towards resisting the antibiotic cefotaxime in an *E. coli* strain with a high mistranslation rate [39]. Other experiments have demonstrated that substitutions at residues 69, 130, 165, 244, 275, and 276 from *bla*_TEM-1_ are all believed to play an exclusive role in inhibitor resistance [40,41], and most of them are found at a high frequency in the dataset of clinical isolates. However, it is worth noting that the TV1 substitution with respect to TV2 occurs at residue 116. The *bla*_CTX-M-1__CV1 and *bla*_CTX-M-1__CV2 subvariants were present in the same species as *bla*_TEM-1__TV2. Previous studies found five key substitutions among CTX-M-1 variants: Val80Ala, Asp117Asn, Ser143Ala, Asp242Gly, and Asn289Asp [42], with the first one being consistent with the variants found in this study.

This is also the first report for the detection of *bla*_OXA-181_ from a non-hospital environment in Ecuador. Since broad-spectrum cephalosporins and carbapenems are both weakly hydrolyzed by OXA-48 and OXA-181 carbapenemases, elevated minimum inhibitory concentrations (MICs) for those drugs may not be apparent in traditional phenotypic tests [43]. Therefore, isolates harboring OXA-48 and OXA-181 may go undetected in routine laboratory settings, complicating treatment options. In addition, they have a high dissemination rate due to transferable plasmids, making them an important cause of a wide range of infections, both in community and healthcare settings [43]. Thus, molecular detection using eDNA could be a sensitive tool for OXA48/181 surveillance. During our search in 2022, the subvariant OV1 corresponded to a single 100% match in GenBank; thus, OV1 may be a new circulating variant that should require further analysis in future isolates from QPT.

*Staphylococcus* bacteria and *mecA* gene overlap throughout QTP stations. These findings were consistent with similar studies on public facilities [44]. Interestingly, the Coagulase-positive staphylococci (CoPS), widely reported as *mecA* carriers, including *S. aureus* [44], represented just 1% of the sequences. However, the Coagulase-negative Staphylococci (CoNS) was considerably more abundant, with nine species that comprise 7.4% of QPT bacteria identified (Figure 2; Appendix A). Despite being formerly categorized as lesser or nonpathogenic, CoNS today comprises a significant group of nosocomial pathogens [45], including *S. epidermidis* and *S. haemolyticus*, both of which were found in QPT (Figure 2; Appendix A). The widespread presence of the *mecA* gene in QTP may be a warning sign for the expansion and selection of *mecA* into other commensal and environmental bacteria, including CoNS species [46].

The bacterial diversity in QPT showed the typical bacterial classes found in environmental and anthropogenic surfaces, where Alphaproteobacteria, Gammaproteobacteria, Bacilli, and Actinobacteria are the most abundant [47]. Three species found in QPT (*S. aureus*, *E. coli*, and *S. pneumoniae*) along with *K. pneumoniae* and *P. aeruginosa* were the etiological agent for 54.9% of deaths among 33 investigated bacteria in a systematic analysis for the Global Burden of Disease in 2019 [48]. It is not possible to properly differentiate the amount *E. coli* strains through 16S rDNA barcoding; thus, this species might play different roles in the sampled environment of the QPT, from probiotic, to pathogen, to commensal [49]. However, *E. coli* has been identified as one of the most significant sources of resistance genes that may be to blame for treatment failures in both human and veterinary medicine [43]. *S. aureus* (S1, S5, and S9) and *S. pneumoniae* (S5, S15) were found in the South Line along with β-lactam resistance genes *bla*_TEM_, *mecA* except in S5. Meanwhile, *E. coli* was present in South and Central lines stations (S11, S13, S16, and C4) and overlapped β-lactam and carbapenem resistance-genes *bla*_TEM_, *bla*_CTX-M-1_ and *bla*_OXA-181_ (Figure 1). 

Pathogen and resistant gene coincidences suggest that the dispersion of MDR bacteria is continuous and diverse in public transport systems with a high density of users. These findings agree with similar studies in public transport in big cities such as London [50]; Lisbon [51]; Guangzhou [52], Tokyo, and Niigata [53], where MDR bacteria can often be isolated from frequently touched surfaces of public transportation serving both hospital and community routes.

Other opportunistic and noncommon pathogens were widely identified in QPT. These species are usually significant in hospital settings, causing infections in immunocompromised patients. The Gammaproteobacteria phylum showed several species that exemplify this phenomenon, such as *Pantoea aglomerans*, *P. calida* [54], *Pseudomonas mendocina* [55], and *Acinetobacter radioresistens* [56], all agents of nosocomial bacteremia. Although opportunistic bacteria rarely cause infections, the high abundance reported in our study suggests a permanent risk of infection for the user community of the QPT and input for nosocomial environments.

The current SARS-CoV-2 pandemic has reinforced the world’s ability to rapidly diagnose infectious disease outbreaks to take suitable epidemiological measures and minimize negative impacts [57]. On top of this, the silent pandemic of antimicrobial resistance (AMR) presents outstanding challenges related to its evolution and propagation. Understanding the resistance genes and bacteria present in public transport systems, where thousands of people share reduced spaces, is a valuable contribution to the creation of public health strategies at the service of municipalities and governments.

## 5. Conclusions

ESBL genes, *bla*_TEM-1_ and *bla*_CTX-M-1_, and MBL genes, *bla*_OXA-181_ and *mecA*, were detected along QPT stations. This is the first report of *bla*_OXA-181_ in an Ecuadorian environmental sample. Our results suggest that CoNS bacteria might have a role as *mecA* resistance reservoirs, and other environmental Gram-negative bacteria might act similarly for *bla*_TEM-1_, *bla*_CTX-M-1_, and *bla*_OXA-181_. The overlapping presence of these resistance genes with opportunistic pathogens (classified as critical and high priority by WHO) in QPT should be considered as potential health hazards and included in future sanitary plans for the QPT. Our work contributes to understanding the prevalence of antibiotic resistance in public transport systems and is a source of information for health planners working on the etiological agents circulating among the inhabitants of the Quito metropolitan area.

## Figures and Tables

**Figure 1 ijerph-20-01900-f001:**
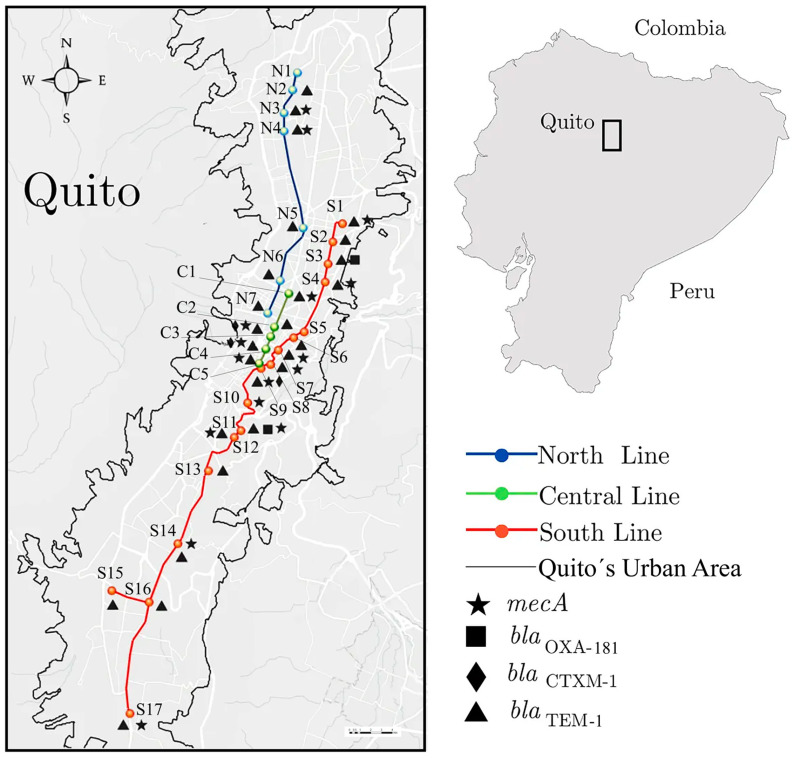
Map of Quito’s Public Transport (QTP) showing North, Central, and South lines. Black icons show the presence of *bla*_TEM-1_, *bla*_CTX-M1_, *bla*_OXA-48_, and *mecA* genes found in each station from eDNA.

**Figure 2 ijerph-20-01900-f002:**
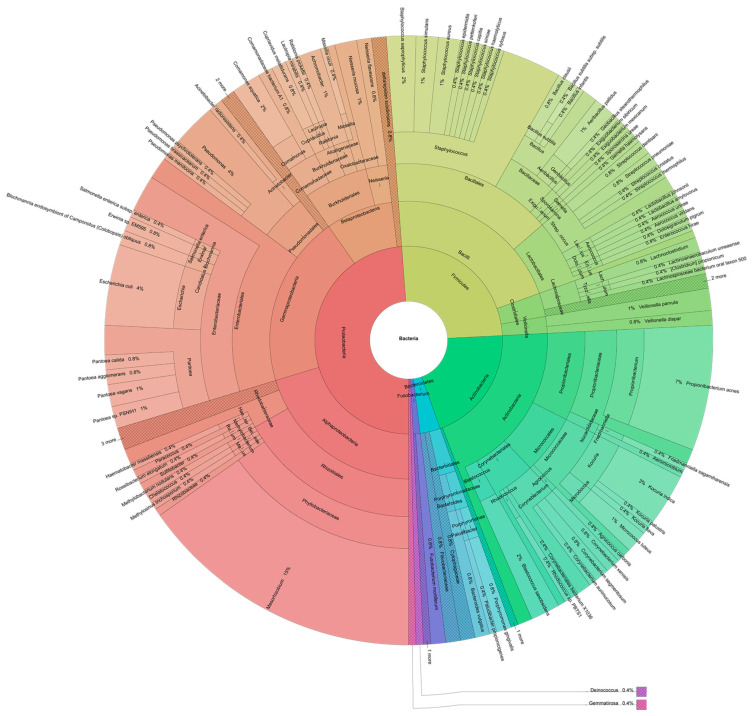
Kronas plot of the relative abundance of species detected in eDNA samples from 16S rDNA in the Quito Public Transport (QTP) during Mach 2022. Data set available at: https://usegalaxy.org.au/datasets/a6e389a98c2d1678a9b3a242082abea8/display/?preview=True&dataset=0&node=1&collapse=true&color=false&depth=8&font=10&key=true (accessed on 5 December 2022).

**Table 1 ijerph-20-01900-t001:** List of primers used for molecular identification of β-lactam resistance genes from eDNA of QPT surfaces.

Gene	Primer’s Name	Sequence (5′-3′)	PCR Product (bp)	Annealing (°C)	Reference
16S rDNA	8F	AGAGTTTGATCCTGGCTCAG	527	57.4	[22]
	534R	ATTACCGCGGCTGCTGG			
*bla* _TEM_	TEM-410F	GGTCGCCGCATACACTATTCTC	372	60	[20]
	TEM-781R	TTTATCCGCCTCCATCCAGTC			
*bla* _SHV_	SHV-287F	CCAGCAGGATCTGGTGGACTA	231		
	SHV-517R	CCGGGAAGCGCCTCAT			
*bla* _CTXM-1_	ctxm1-115F	GAATTAGAGCGGCAGTCGGG	588		
	ctxm1-702R	CACAACCCAGGAAGCAGGC			
*bla* _CTXM-2_	ctxm2-39F	GATGGCGACGCTACCCC	107		
	ctxm2-145R	CAAGCCGACCTCCCGAAC			
*bla* _CTXM-9_	ctxm9-16F	GTGCAACGGATGATGTTCGC	475		
	ctxm9-490R	GAAACGTCTCATCGCCGATC			
*bla* _CTXM-8/25_	ctxm8g25g-533F	GCGACCCGCGCGATAC	186		
	ctxm8g25g-718R	TGCCGGTTTTATCCCCG			
*bla* _KPC_	KPCfw	CGTCTAGTTCTGCTGTCTTG	798	55	[23]
	KPCrv	CTTGTCATCCTTGTTAGGCG			
*bla* _VIM_	VIMfw	GATGGTGTTTGGTCGCATA	390		
	VIMrv	CGAATGCGCAGCACCAG			
*bla* _NDM_	NDMfw	GGTTTGGCGATCTGGTTTTC	621		
	NDMrv	CGGAATGGCTCATCACGATC			
*bla* _OXA-48/181_	OXA48fw	GCGTGGTTAAGGATGAACAC	438		
	OXA48rv	CATCAAGTTCAACCCAACCG			
*mecA*	mecA147-F	GTGAAGATATACCAAGTGATT	147		[24]
	mecA147-R	ATGCGCTATAGATTGAAAGGAT			
	mecDrv	CTCCCATCTTTTCTCCATCCT			

## Data Availability

Sequences were submitted to the nucleotide database at NCBI and are available under the following accession numbers: antibiotic resistance partial-genes blaCTXM-1V1-V2: OP846056-7; blaTEM-1V1V2: OP846058-9; mecA OP846060; and blaOXA-48V1-V2: OP846061-2. 16S rDNA sequences: OP925023–OP925296 (Appendix A).

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
