# Peer review of "Beta-Lactam Antibiotic Resistance Genes in the Microbiome of the Public Transport System of Quito, Ecuador"

_ijerph, 2023, doi:10.3390/ijerph20031900_

Round 1
Reviewer 1 Report
Title: First screening of Beta-lactam Antibiotic Resistance Genes and bacterial diversity in the Public Transport System of Quito, Ecuador
This manuscript is well-written by the authors. I do believe that if they can improve the manuscripts following all comments. It might have a chance to publish in the journal.
Comments
1. Topic: Please rewrite
I have suggested
" Multi-drug resistance and beta-lactam resistant genes in microbiome circulating in Quito’s Public Transport system of Quito, Ecuador "
Abstract
2. Does the author identify the species of the bacteria? Please describe
3. Which bacterium is the most prevalence in microbiome circulating in Quito’s Public Transport?
4. Which resistance gene is the most prevalence in microbiome? Please present
5. It is better if the author writes some sentences using passive voice.
Introduction
6. Line 45: the reference style is different, compared with other references.
7. Line: 61. The words “TEM, SHV, and CTX-M types.” the author should write the full name of each word, and present the abbreviation in the bracket such as penicillin (PEN).
8. Line 63-65: the reference style is different, compared with other references.
9. Line 66: please check the word “β--lactamases”. The correct is β--lactamases or β-lactamases.
10. Line 93, please modify the sentence.
Materials and methods
11. Study area and sample collection: how many areas of each swab? For example, 10 x 10 cm2
12. Line 126: please delete the word “by the authors”
13. Line 127: please delete the word “between March and August 2022” (in Table 1)
14. Do the authors perform the statistical analysis for analyzing the data or results? If yes, please describe the statistical analysis in the materials and methods.
Results
15. Line 148-149: please remove and add in the first sentence of “QTP β-lactam resistance genes screening”
16. Line 160: the PCR products
17. Line 193: the words “Bacteroidetes and Fusobacterium” should be italic.
18. Line 197: Fifteen percent
19. Line 204 and 207: showed
Discussion
20. Line 246: please carefully check that this research is the first report on blaOXA-181 detection.
21. Line 257: The authors have repeated the results, please modify.
22. Line 270: please use the word “S. aureus”
23. Please add the information of the discussion. Try to compare the results (the author’s hypothesis) with other finding by other researchers.
24. Please cite the update references.
Conclusion
25. Some sentences in the conclusion should be written using the past tense, please check.
References.
26. There are many references. In general, there are 30-40 references for each manuscript (research articles). Please delete some unnecessary references. Please cite the updated references.
Author Response
Title: First screening of Beta-lactam Antibiotic Resistance Genes and bacterial diversity in the Public Transport System of Quito, Ecuador
This manuscript is well-written by the authors. I do believe that if they can improve the manuscripts following all comments. It might have a chance to publish in the journal.
Comments
- Topic: Please rewrite
I have suggested
" Multi-drug resistance Beta-lactam resistant genes in microbiome circulating in the Public Transport system of Quito, Ecuador "
We accept the suggestion with a small modification: “Beta-lactam resistant genes in microbiome present in the Public Transport system of Quito, Ecuador”.
Abstract
- Does the author identify the species of the bacteria? Please describe
We addressed the process of species identification in line 133 of the methods section and line 204 of the results section. We explain how we use a region of the 16S ribosomal gene for this purpose.
- Which bacterium is the most prevalence in microbiome circulating in Quito’s Public Transport?
Our study is focused on the prevalence of genes for antibiotic resistance and not on the prevalence of specific pathogens of bacterial taxa. We expect to release information on bacterial ecology and taxa abundance in the public transport system on a second manuscript. However, we do refer the the most abundant phylla, Mesorhizobium, in the results section.
- Which resistance gene is the most prevalence in microbiome? Please present
We added “ESBL genes blaTEM-1 and blaCTX-M-1 and MBL genes blaOXA-181 and mecA were detected along QPT stations being blaTEM the most widely spread” in line 29.
- It is better if the author writes some sentences using passive voice.
We would like to maintain the use of active voice. A recent guideline provided by an important scientific publishing editorial, encourages the use of active voice in scientific manuscripts. Please refer to the following online reference: https://scientific-publishing.webshop.elsevier.com/manuscript-review/using-active-and-passive-voices-academic-writing/
Introduction
- Line 45: the reference style is different, compared with other references.
Done
- Line: 61. The words “TEM, SHV, and CTX-M types.” the author should write the full name of each word and present the abbreviation in the bracket such as penicillin (PEN).
We added: “These enzymes stimulate hydrolysis of the β-lactam ring and thereby inhibit these antibiotics [6]. Among ESBLs, the most widespread and clinically relevant are the class Temoneira (TEM), Sulfhydril (SHV), and Cephotaxime Munich (CTX-M) types.”
- Line 63-65: the reference style is different, compared with other references.
Done.
- Line 66: please check the word “β--lactamases”. The correct is β--lactamases or β-lactamases.
Done
- Line 93, please modify the sentence.
We changed to: One of the strategies for MDR surveillance is the detection of antibiotic resistance genes in pathogenic and non-pathogenic microorganisms by amplifying and sequencing specific genes directly from environmental DNA (eDNA).
Materials and methods
- Study area and sample collection: how many areas of each swab? For example, 10 x 10 cm2
We added: “Bacteria were collected as described by Rawlinson et al by rubbering three sterilized cotton swabs (previously soaked in PBS 1X, pH 7.4) on the entire facing surface (ca. 20 cm2) of entry and exit turnstiles from 29 stations in March 2022.”
- Line 126: please delete the word “by the authors”
Done
- Line 127: please delete the word “between March and August 2022” (in Table 1)
Done
- Do the authors perform statistical analysis for analyzing the data or results? If yes, please describe the statistical analysis in the materials and methods.
This work is a first step to creating molecular tools that can be used for low-cost routine monitoring in the future. For this reason, this study focused on the detection of resistance genes and circulating bacteria without addressing biodiversity or other types of analysis that require the use of statistics.
Results
- Line 148-149: please remove and add in the first sentence of “QTP β-lactam resistance genes screening”
Done
- Line 160: the PCR products
Done
- Line 193: the words “Bacteroidetes and Fusobacterium” should be italic.
Bacteroidetes and Fusobacterium are phylla, thus are not written in italics, Nevertheless, thanks to your observation we realisez that fusobacterium phylla has been changed to Fusobacteriota. Hence, we chanched so in the manuscript.
- Line 197: Fifteen percent
Done
- Line 204 and 207: showed
Done
Discussion
- Line 246: please carefully check that this research is the first report on blaOXA-181detection.
We have re-reviewed the oxa reports in Ecuador and have not found any environmental reports. In any case, we have changed the sentence so as not to be blunt: “This might be the first report for blaOXA-181 detection from non-hospital settings in Ecuador”
- Line 257: The authors have repeated the results, please modify.
We rewrite lines 256 erasing the results: “Staphylococcus bacteria and mecA gene overlaps throughout QTP stations.”
- Line 270: please use the word “S. aureus”
Done
- Please add the information of the discussion. Try to compare the results (the author’s hypothesis) with other finding by other researchers.
We added: S. aureus (S1, S5, and S9) and S. pneumoniae (S5, S15) were found in the South Line along with β-lactam resistance genes blaTEM, mecA except in S5. Meanwhile, E. coli was present in South and Central lines stations (S11, S13, S16, and C4) and overlapped β-lactam and carbapenem resistance-genes blaTEM, blaCTX-M-1 and blaOXA-181 (Figure 1).
Pathogen and resistant gene coincidences suggest that the dispersion of MDR bacteria is continuous and diverse in public transport systems with a high density of users. These findings agree with similar studies in public transport in big cities such as London [50]; Lisbon [51]; Guangzhou [52], Tokyo, and Niigata [53], where MDR bacteria can often be isolated from frequently touched surfaces of public transportation serving both hospital and community routes.
- Please cite the update references.
Otter, J.A.; French, G.L. Bacterial Contamination on Touch Surfaces in the Public Transport System and in Public Areas of a Hospital in London. Lett Appl Microbiol 2009, 49, 803–805, doi:10.1111/j.1472-765X.2009.02728.x.
Conceição, T.; Diamantino, F.; Coelho, C.; de Lencastre, H.; Aires-de-Sousa, M. Contamination of Public Buses with MRSA in Lisbon, Portugal: A Possible Transmission Route of Major MRSA Clones within the Community. PLoS One 2013, 8, 1–6, doi:10.1371/journal.pone.0077812.
Peng, Y.; Ou, Q.; Lin, D.; Xu, P.; Li, Y.; Ye, X.; Zhou, J.; Yao, Z. Metro System in Guangzhou as a Hazardous Reservoir of Methicillin-Resistant Staphylococci: Findings from a Point-Prevalence Molecular Epidemiologic Study. Sci Rep 2015, 5, 8–12, doi:10.1038/srep16087.
Iwao, Y.; Yabe, S.; Takano, T.; Higuchi, W.; Nishiyama, A.; Yamamoto, T. Isolation and Molecular Characterization of Methicillin-Resistant Staphylococcus Aureus from Public Transport. Microbiol Immunol 2012, 56, 76–82, doi:10.1111/j.1348-0421.2011.00397.x.
Conclusion
- Some sentences in the conclusion should be written using the past tense, please check.
Done
References.
- There are many references. In general, there are 30-40 references for each manuscript (research articles). Please delete some unnecessaryreferences. Please cite the updated references
We deleted some old references and added new ones in discussion (see suggestion number 23). In the end, we got 57 references. Since the mdpi format does not limit the number of references, we take this opportunity to name all the previous works that can help the reader to deepen the topic. We have reviewed the rest of references in the manuscript and made an effort to use strictly those that are essential to the manuscript.

Reviewer 2 Report
See attachment.

Author Response
- Since the authors detected 16S sequences of bacteria belonging to WHO priority list for antimicrobial resistance (to beta-lactams), including E. coli, Acinetobacter baumannii, Klebsiella pneumonia, and Staphylococcus aureus, it would be interesting to know how these bacteria are distributed along the QPT; are there stations where they are more prevalent?
We added: “S. aureus (S1, S5, and S9) and S. pneumoniae (S5, S15) were found in the South Line overlapping β-lactam resistance genes blaTEM, mecA except in S5. Meanwhile, E. coli was present in South and Central lines (S11, S13, S16, and C4) and overlapped β-lactam and carbapenem resistance-genes blaTEM, blaCTX-M-1 and blaOXA-181.”
- Did the above mentioned species significantly overlap with the beta-lactamase genes in the whole QPT or in certain stations?
We added: “S. aureus (S1, S5, and S9) and S. pneumoniae (S5, S15) were found in the South Line overlapping β-lactam resistance genes blaTEM, mecA except in S5. Meanwhile, E. coli was present in South and Central lines (S11, S13, S16, and C4) and overlapped β-lactam and carbapenem resistance-genes blaTEM, blaCTX-M-1 and blaOXA-181.”
- Results section, line153: according to Table S1, blaTEM gene was detected in 26 stations and not 27.
That is true. We changed the sentece in the main text: “The most widely spread resistance gene was blaTEM, detected in 26 stations,…"

Reviewer 3 Report
Please see the attached file

Author Response
Reviewer3
This manuscript aimed to screen beta-lactam antibiotic resistance genes and bacterial diversity in the public transport system in Ecuador. There are few questions since they are not mentioned in the manuscript.
- It is questioned whether the PCR results can be reliable.
The independent effect of PCR reactions on individual DNA samples is a well-known source of bias for studies of bacterial community structure (e. g. abundance and diversity), particularly in those cases where high-throughput sequencing is used. However, our study applies a specific amplification process to detect the presence of individual markers for antimicrobial resistance genes. In the case of the sequencing process, we apply a Sanger technique, which reduces the possible bias introduced with high-throughput sequencing.
The PCR conditions were standardized based on the literature:
TEM1/SHV/CTX-M: Yamaguchi, T.; Kawahara, R.; Harada, K.; Teruya, S.; Nakayama, T.; Motooka, D.; Nakamura, S.; do Nguyen, P.; Kumeda, Y.; Dang, C. van; et al. The Presence of Colistin Resistance Gene Mcr-1 and -3 in ESBL Producing Escherichia Coli Isolated from Food in Ho Chi Minh City, Vietnam. FEMS Microbiol Lett 2018, 365.
KPC/VIM/NDM/OXA: Poirel, L.; Walsh, T.R.; Cuvillier, V.; Nordmann, P. Multiplex PCR for Detection of Acquired Carbapenemase Genes. Diagn Microbiol Infect Dis 2011, 70, 119–123, doi:10.1016/J.DIAGMICROBIO.2010.12.002.
MecA: Zhang, K.; McClure, J.A.; Elsayed, S.; Louie, T.; Conly, J.M. Novel Multiplex PCR Assay for Characterization and Concomitant Subtyping of Staphylococcal Cassette Chromosome Mec Types I to V in Methicillin-Resistant Staphylococcus Aureus. J Clin Microbiol 2005, 43, 5026–5033, doi:10.1128/JCM.43.10.5026-5033.2005.
Nevertheless, we did not have MecC-D positive controls, so we removed those two resistances from the main text as follows: “Four out of eleven genes for β-lactam antibiotic resistance were detected: blaTEM, mecA, blaCTX-M-1, and blaOXA181. The most widely spread resistance gene was blaTEM, detected in 26 stations, followed by mecA in 16 stations, blaCTX-M-1 in three, and blaOXA181 in two. Meanwhile, blaSHV, blaCTX-M2, blaCTX-M8/25, blaCTX-M9, blaKPC, blaVIM, and blaNDM, did not amplify in any sample (Figure 1; Table S1).”
- “9 of 13 resistance genes are not detected” How many times have you done for PCR in all stains? a. What are the bacterial stRains of positive controls for PCR in every single gene?
- Our study uses an experimental approach based on the presence or absence of particular markers for antimicrobial resistance genes; in this sense, our study presents a baseline to develop more complex and robust surveys through a longitudinal design that may allow to develop mathematical models on the presence, prevalence and intensity on the distribution of these particular genes.
Since it is a cross-sectional study, we took three samples from every station just once. Every PCR was standardized using the positive control and double check. Nevertheless, it might be possible that these genes were not present in QPT or the sawed area at this time. Thus, in results we did not affirm that these genes are not present in QPT, but we were not able to amplify them.
- Do you re-check the primers and the PCR conditions? If yes, please show the positive result of each gene.
We attached a word document with the captios of the positive controls
Every positive result has been sequenced and is available at the GenBank: blaCTXM-1V1-V2: OP846056-7; blaTEM-1V1V2: OP846058-9; mecA OP846060; and blaOXA-48V1-V2: OP846061-2.
- There are some typo errors. Please double check throughout the manuscript.
Done
- Supplement data should be separated from the main manuscript.
- We have considered this recommendation within the provided guidelines of the IJERPH
- Please recheck in-text references and also references at the end in MDPI style
We have made the necessary amendments to the literature cited section.

Reviewer 4 Report
This paper presents a screening of ꞵ-lactam Antibiotic resistance genes and bacterial diversity in the public transport system in Quito, Ecuador. To my knowledge, this is the first approach conducted in Quito public transport system. The work is well conducted, with appropriated new technics. The results are convincing and seem trustable. I guess that it has some original information, worth knowing. However, in Material and methods section, no indication had been made on the number of samples for each station, sampling methods must be detailed.
Author Response
Reviewer4
Open Review
This paper presents a screening of ꞵ-lactam Antibiotic resistance genes and bacterial diversity in the public transport system in Quito, Ecuador. To my knowledge, this is the first approach conducted in Quito public transport system. The work is well conducted, with appropriated new technics. The results are convincing and seem trustable. I guess that it has some original information, worth knowing. However, in Material and methods section, no indication had been made on the number of samples for each station, sampling methods must be detailed.
We added: “Bacteria were collected as described by Rawlinson et al by rubbering three sterilized cotton swabs (previously soaked in PBS 1X, pH 7.4) on ca. 20 cm2 surface of entry and exit turnstiles from 29 stations in March 2022. The three swabs from the same station were merged in 1.5mL microtube obtaining 29 samples for further DNA extraction.”

Round 2
Reviewer 2 Report
The authors provided satisfactory responses to all the questions and comments. I recommend this manuscript to be accepted for publication.
Reviewer 3 Report
This manuscript shows a fair information. After amendment, it has a chance to publish in the journal.